# Migrants and Service Providers’ Perspectives of Barriers to Accessing Mental Health Services in South Australia: A Case of African Migrants with a Refugee Background in South Australia

**DOI:** 10.3390/ijerph18178906

**Published:** 2021-08-24

**Authors:** Nelsensius Klau Fauk, Anna Ziersch, Hailay Gesesew, Paul Ward, Erin Green, Enaam Oudih, Roheena Tahir, Lillian Mwanri

**Affiliations:** 1College of Medicine and Public Health, Flinders University, Adelaide, SA 5042, Australia; nelsensius.fauk@flinders.edu.au (N.K.F.); Anna.ziersch@flinders.edu.au (A.Z.); Hailay.gesesew@flinders.edu.au (H.G.); paul.ward@flinders.edu.au (P.W.); Erin.green@flinders.edu.au (E.G.); roheena.tahir@flinders.edu.au (R.T.); 2Institute of Resource Governance and Social Change, Jl. R. W. Monginsidi II, No. 2, Kupang 85221, Indonesia; 3Relationships Australia, Adelaide, SA 5007, Australia; E.Oudih@rasa.org.au

**Keywords:** mental healthcare services, mental health problems, barriers, African migrants, Australia

## Abstract

International mobility has increased steadily in recent times, bringing along a myriad of health, social and health system challenges to migrants themselves and the host nations. Mental health issues have been identified as a significant problem among migrants, with poor accessibility and underutilisation of the available mental health services (MHSs) repeatedly reported, including in Australia. Using a qualitative inquiry and one-on-one in-depth interviews, this study explored perspectives of African migrants and service providers on barriers to accessing MHSs among African migrants in South Australia. The data collection took place during the COVID-19 pandemic with lockdown and other measures to combat the pandemic restricting face to face meetings with potential participants. Online platforms including Zoom and/or WhatsApp video calls were used to interview 20 African migrants and 10 service providers. Participants were recruited from community groups and/or associations, and organisations providing services for migrants and/or refugees in South Australia using the snowball sampling technique. Thematic framework analysis was used to guide the data analysis. Key themes centred on personal factors (health literacy including knowledge and the understanding of the health system, and poor financial condition), structural factors related to difficulties in navigating the complexity of the health system and a lack of culturally aware service provision, sociocultural and religious factors, mental health stigma and discrimination. The findings provide an insight into the experiences of African migrants of service provision to them and offer suggestions on how to improve these migrants’ mental health outcomes in Australia. Overcoming barriers to accessing mental health services would need a wide range of strategies including education on mental health, recognising variations in cultures for effective service provision, and addressing mental health stigma and discrimination which strongly deter service access by these migrants. These strategies will facilitate help-seeking behaviours as well as effective provision of culturally safe MHSs and improvement in access to MHSs among African migrants.

## 1. Introduction

Australia has a significant migrant population, with arrivals from many developing settings including from Asia and Africa [1,2]. According to the 2020 Australian Bureau of Statistics report, there were over 7.6 million migrants living in Australia, of whom 380,000 were from African countries [3]. During the period between 2009 and 2018, Australia resettled 180,788 people with a refugee background, accounting for about 0.89% of an estimated 20.3 million refugees globally [4]. For African migrants, the majority have migrated to Australia as refugees due to a wide range of factors including civil conflicts and/or war in their home countries [5,6]. They may have had negative experiences during relocation, especially when residing in refugee camps, and resettlement challenges make it more complex to settle in host countries [6,7,8,9]. In Australia, African migrants have been reported to experience a disproportionate burden of health problems related to postmigration resettlement, such as trauma, separation from family and peer networks, difficulties in social adaptation and adjustment to new systems, unemployment, and challenges accessing housing and education [5,7,10,11,12].

These settlement issues facing African migrants, particularly those arriving with a refugee background can contribute to mental health problems (MHPs) such as depression, schizophrenia, and stress conditions [7,13,14,15]. It is also well known that migrants especially those with a refugee background are at a higher risk of common mental disorders, including ten times higher rates of post-traumatic stress disorders (PTSD) and depression [1], further affecting resettlement processes and engagement with mental healthcare services (MHSs) in Australia [5,9,16,17]. The 2020 Australia’s Health report showed that one in five Australians (20.1%) experienced a mental health condition in 2017–2018, an increase from 17.5% in 2014–2015 [18]. The current COVID-19 pandemic and associated public health measures have been reported to increase and exacerbate MHPs such as anxiety, depression, and stress within migrant populations in Australia [19,20,21,22].

The availability of MHSs is considered a human rights issue, but globally, there are recognised limitations to accessing these services [23]. In many developed countries, including Australia, migrant populations are often considered homogenous, and significant attributes such as cultures and differentials in backgrounds, especially related to the countries of origin and circumstances of migration, are not taken into consideration when planning and implementing services [24]. In Australia, there are also few Culturally and Linguistically Diverse (CALD) specific services and a fragmented health system can make it a challenge for migrants to access services [25,26,27,28]. Although the prevalence of MHPs such as PTSD in refugees is known to be higher than the general population in resettlement countries [29,30], the pattern of mental health help-seeking behaviour is also recognised to be lower in these communities [28,31]. Evidence demonstrates that the place of birth influences the likelihood of seeking MHSs or treatment [28,32]. For instance, Greek migrants are more likely to access MHSs than the Australian-born population, though, the opposite has been reported for Southeast Asian or Irish migrants in Australia [32].

Although the Australian government has issued the Australian Mental Health Plan that aims to provide general MHSs for all Australians [33], the services are generally poorly accessed by CALD migrants. Available evidence suggests that MHSs are underutilised among CALD migrants in Australia compared to the general Australian community, due to a wide range of barriers [28,30,34,35,36]. These include policy-level barriers where the specific needs of CALD communities are not acknowledged [28,37]; mental health stigma and discrimination, where patients may fear to be seen accessing the services; poor health literacy including unfamiliarity with the services; fear of breach of confidentiality; and other issues related to the quality of care and services, such as long waiting lists, and affordability [28,34,37]. With regard to accessing MHSs, the current COVID-19 pandemic is also reported as a barrier to the access to MHSs among migrants and refugee populations in Australia and many other settings due to worry, stress and fear of exposure to COVID-19 infection and is considered to be exacerbating the pre-existing barriers to accessing MHSs among these populations [18,19,20,21,38]. Despite a range of barriers reported in previous studies, there is a paucity of evidence on sociocultural and religious factors influencing access to MHSs among CALD migrants, especially in Australia.

In order to identify more specific needs and improve access to and utilisation of MHSs in Australia, the current study sought to develop an understanding of barriers to accessing the services in migrants, primarily from Sub-Saharan African (SSA).

### Theoretical Framework

To guide the analysis and discussion of the current findings, the access to healthcare services conceptual framework was applied [39]. This framework suggests that, for patients to access healthcare services, there should be interactions between dimensions of accessibility of these services and the corresponding abilities of people in need of health care. Dimensions of accessibility of services include availability, approachability, affordability, appropriateness, and acceptability. Availability refers to the existence of the services and of healthcare professionals with sufficient capacities to produce or deliver the services to people with health needs. Approachability refers to the concept that healthcare services should make themselves known to people within various social and geographical groups and communities, which enables people in need to identify their existence and access them. Affordability reflects the direct prices of the services and whether the prices of the services are affordable to people with health needs. Appropriateness relates to the fit between the services and the needs of people accessing the services. Acceptability relates to cultural and social factors, such as the sex and social groups of service providers, beliefs associated with certain medicine, etc., which determine the possibility for people to accept the services [39] This framework also acknowledges that the accessibility of healthcare services should be supported by the abilities of people with health needs, including the ability to reach the services or healthcare facilities, a perceived need for the services, affordability of the services and engagement in the services, determined by individual motivations in seeking the services.

Guided by the access to healthcare framework [39], this study aims to explore personal, structural, sociocultural and religious barriers to accessing MHSs among African migrants in South Australia. Understanding these barriers is necessary for governments and policy makers to address them at policy and practical level through programs or interventions that improve African migrants’ access to these services and mental health outcomes.

## 2. Materials and Methods

The consolidated criteria for reporting qualitative studies (COREQ) checklist was used to guide the report of the methods section (Appendix A) [40]. The checklist suggests 32 items that should be addressed to support the explicit and comprehensive reporting of qualitative studies.

### 2.1. Study Design and Sampling

This study employed a qualitative design, using one-on-one in-depth online (WhatsApp or Zoom) interviews with 20 African migrants and ten service providers in South Australia. The use of qualitative design enabled the researchers to collect data to explore participants’ perceptions, understanding, experience and interpretation of factors that influenced accessibility to MHSs [39,40,41].

Participants were recruited using the snowball sampling technique. The recruitment started with the dissemination of study information packs, containing contact details of the field researcher, to potential participants through various African community groups or associations, and organisations providing services for migrants or refugees in South Australia. The researcher (EG) has formal training in public health and qualitative research methods. Initial participants who called, sent a text message or emailed to confirm their participation were recruited for an interview. Interviewees were asked to help further disseminate the information packs to their eligible friends, colleagues and networks who might be willing to participate in the study. The recruitment process was recursive and took three months. Participants were eligible for recruitment if they were an African migrant aged 18 years old and above (community participants), or a person working for an organisation/institution providing services for migrants and/or refugees (for service providers). None of the potential participants who had confirmed to wish to be interviewed withdrew their participation before or during the interviews. Thirty participants, comprising 20 African migrants and 10 service providers met the inclusion criteria and participated in the study.

### 2.2. Data Collection Procedure

In-depth interviews were intended to be conducted face-to-face but, due to the Covid-19 protocols and restrictions, data collection was conducted online or by phone. Interviews were conducted in English and audio recorded digitally. Additionally, notes were taken during the interview process. Interviews focused on several key areas including: (i) participants’ knowledge or understanding about MHPs and MHSs, (ii) participants’ experience of procedures related to access to MHSs, (iii) how participants reached healthcare facilities providing MHSs, and (iv) participants’ economic or financial conditions, and how these conditions influenced their access to MHSs. Additionally, the study explored African communities’ perceptions about MHPs, the influence of those perceptions on people with MHPs in accessing MHSs, and the decision-making processes within African families and their influence on their access to MHSs. We also explored what supports or services African migrants usually accessed to help with coping with any MHPs, and whether there were any other factors that would influence their access to MHSs (see interview guide, Appendix B). None of the participants were known to the researcher prior to the interviews. The recruitment of the participants and interviews ceased once data saturation had been reached. This was reflected in the information or answers to research questions by the last few participants, which were similar to those of previous participants. No repeated interviews were conducted. Each participant was offered an opportunity to read and correct the interview transcript at the end of the project, but none requested to do so.

### 2.3. Data Analysis

Audio recordings were transcribed verbatim and analysed thematically using the framework approach by Ritchie and Spencer [42], undertaken manually using Microsoft word in order to group themes. The framework approach comprises five steps of qualitative data analysis, including familiarisation with the data, identification of a thematic framework, indexing the data, charting the data and mapping and interpretation of the data. Familiarisation with the data was performed by reading each transcript repeatedly, breaking down the data into small chunks, highlighting and making comments on the data. Identification of a thematic framework was carried out through identifying and writing down the recurrent and emerging key issues and concepts, and a thematic framework was applied to develop a coding scheme for the data. The thematic framework was identified and developed deductively based on concepts in the access to healthcare framework presented in the previous section and on prior knowledge, and inductively based on themes that emerged from the data. Indexing the data was conducted through the coding process, starting by creating open codes to the data extracts of each individual transcript, which resulted in a long list of codes. This was followed by close coding to identify similar codes and reduce the number codes to a manageable number, and different codes that seemed to form a theme or sub-theme were then grouped together. Finally, a shortlist of four overarching themes was reached: personal barriers, structural or health system-related barriers, sociocultural barriers and religious barriers. The next step was creating a chart for the data which was done by arranging appropriate thematic references in a summary chart for further comparison of the data across interviews and within each interview. Finally, mapping and interpretation of the data took place. Application of this framework analysis helped the management of these qualitative data in a structured way and enhanced rigour and transparency of the analytical process [42,43,44,45]. Transcript coding, charting and interpretations were primarily performed by two researchers (LM and NF), although team-based analysis and discussion were also performed at regular meetings. Initially, each researcher undertook an independent analysis of themes and the results of the analysis were brought into discussions at the regular meetings, and finally research team decisions were made about the validity of the final themes and interpretation. Themes identified were personal barriers, health system-related barriers, sociocultural barriers and religious or spiritual related perceptions of MPHs, regarding access to MHSs (see Figure 1).

### 2.4. Ethical Consideration

The study ethics approval and its modification to allow this online data collection approach was obtained from the (then) Social and Behavioural Research Ethics Committee (SBREC) of Flinders University. Immediately prior to the interviews, each participant was informed about the aim of the study and the procedures. Participants were advised that interviews would be conducted and recorded using Zoom or WhatsApp digital platforms. As their participation in this study was voluntary, they were advised of their rights to withdraw their participation prior to or during the interviews without any consequences. The participants were also assured that the collected information would be treated confidentially and anonymously (by de-identifying the data) to prevent the possibility of linking individuals to the data in the future. Each participant signed an informed consent form to confirm their willingness to participate, and they returned the consent form to the interviewer prior to the commencement of the interviews. All participants were offered the opportunity to use an interpreter, but none chose to do so.

## 3. Results

### 3.1. Sociodemographic Profile of the Participants

A total of 20 African migrants were interviewed in this study and half were male and female respectively, with the age range between 18 and 60 years old. The participants were from eight different African countries (see Table 1). Most participants had lived in Australia for between 11 and 20 years at the time of the interviews, while the rest had lived in Australia for 6 to 10 years and 1 to 5 years. Most of the participants [11] had paid jobs, seven [7] were currently unemployed or actively looking for a job, and two participants were full-time students. The service providers interviewed were five males and five females. They engaged in different roles or positions in their institutions.

The following sections outline key themes drawn from the analysis related to participants’ perspectives of barriers to mental health service access for African migrants.

### 3.2. Personal Barriers

#### 3.2.1. Lack of Knowledge of Mental Health Problems and Services

A lack of knowledge or understanding about MHPs and MHSs was a significant barrier to accessing MHSs among African migrants in South Australia. Most participants, including both male and female participants from across different countries suggested that many African migrants were not aware of, or did not recognise MHPs and MHSs available for them, which seemed to lead to poor access to the services. The quotes from female and male African migrants who have lived in Australia for 11 and 15 years, respectively, reflected such a lack of understanding, which seemed to also indicate limited dissemination of information about this topic within African communities:

*“I think it* [barrier to access MHS] *was also understanding what the services entailed, and really understanding what mental illness is. I mean, I didn’t properly understand mental health and mental illness until I came to university, and I actually started doing psychology as a course. I mean, high school, they talk about mental health and wellbeing a little bit, but they’re not really so focused on actually teaching you what it is and you know, there’s different levels and because you do have a mental illness, it’s impacting your life. I didn’t get much information about this at school or within the community where I live”*(African migrant 8, female, Somalia)

*“The main barrier* [to access to MHSs] *is that people don’t recognise that they are sick or they are suffering mental illness as such, then they are not prepared to access services, because they feel they fit”*(African migrant 14, male, Burundi).

The lack of knowledge and understanding of MHPs and MHSs by African community members was also noted by service providers interviewed, as highlighted below:


*“I think the other barriers (to the access to MHSs) will be—it’s a lack of understanding around that mental health and well-being. …. The African community, I think they have limited understanding around mental illness, especially, based on the clients that I’ve been seeing, I do feel like they have no idea if we talk about mental health wellbeing. And also, I think there also is a lack of belief that mental health or mental illness is really there. …. I think lack of understanding around mental health and, I don’t know, mental illness, it’s I think the key. The key and the main barrier (to the access to MHSs)”*
(Service Provider 3, female)

Both African migrants and service providers described the existence of misconceptions and poor understanding of mental health issues within African communities. The following narratives of an African migrant who came to Australia in 2004 and a service provider illustrate such misconceptions, suggesting that mental health issues were seen as a form of overt ‘craziness’ which would have been demonstrated through behaviours, such as running on the street naked, and behaving obviously ‘abnormally’:


*“I think they really don’t even acknowledge there is a problem. One, because this person is not running on the street naked, because the traditional mental health that is known is for someone who is running on the street naked, that’s what we call that they have mental illness. Anything else doesn’t count. It doesn’t count as being a mental illness. …. I actually didn’t even think it was depression, I just thought it was just something that’s happening, getting emotional or something, but when I look back, I think I had depression. Now that I know, I know that it was depression”*
(African migrant 6, female, Burundi)


*“Normally when people hear the word “mental health” in the community it is mainly for people that are considered—I guess, not normal, and everyone see you as being crazy”*
(Service Provider 5, Female)

#### 3.2.2. Lack of Perceived Benefits of Mental Healthcare Services

Poor perceptions of the benefits of MHSs seemed to hinder African migrants from accessing the services. The western model of service delivery that is currently available in Australia such as counselling services were seen as culturally incongruent with the expectations of African clients and hence their benefits were not acknowledged. The quote below from a female participant from Kenya supports these assertions:


*“Also, another thing (that prevent her from accessing the services), I just feel like talking about it won’t really change, because they will just tell me to stay safe, to look after my children, and it’s always the same thing. …. So that’s also another thing that has made me not to speak to the health services”*
(African migrant 19, female, Kenya)

Another participant from the Democratic Republic of Congo described the services to be ineffective and that she believed that people would access these services to please the service providers:


*“.… You just go there and talk or sing, or they make fun of even relaxation technique and stuff like that. So, things like that, if I’m just coming and I’m just doing things to please you with no thought that it’s going to make a difference, then it’s not even going to have an effect”*
(African migrant 10, female, Democratic Republic of Congo)

The cultural inappropriateness of the service model was also acknowledged by the service providers. The quote below is an example of a narrative from a service provider who had worked with African migrants for several years as a wellbeing coach and reflected on how these migrants perceived MHSs provided to them:


*“I can give a story of a guy that I was speaking to that told me he had been connected with a therapist doing trauma work. …he said, “Look, I’ve done it before. My clinician just asks me questions, questions, questions. And I came in, he asked the same questions in different ways. How is this helping me?”*
(Service Provider 8, Male)

#### 3.2.3. Poor Financial Condition and Transport Issues

Poor financial circumstances and issues around transportation to mental healthcare facilities seemed to have a significant influence on access to MHSs among African migrants. Both community and service provider participants indicated that costs related to the access to MHSs and other family needs challenged access to the services. The following narratives illustrate the complexity of financial issues that African migrants faced at both individual and familial levels and their role in hindering MHS accessibility:


*“If four of my kids have mental illnesses, then I have to pay for four of them to get (access the services)—you know. And with Medicare, I think you get seven—oh, no, ten sessions, and then it’s like okay, well, they’re only getting ten sessions (for free), I have to pay for the next few sessions. And then, that’s very costly. And then, I have private health insurance. …. Why am I putting $100, $200 for one session when my kid could use that for their new school uniform, you know?”*
(African migrant 8, female, Somalia)


*“Financial issue is also a barrier for many of them (African migrants) to pay for transportation or healthcare services. …. Just lately—I have to share quickly something. I took a client to a service for financial issues and when I went there with him, he really was hopeless because he got sent an electricity bill. On hand, he got a letter saying like, okay, if you don’t pay by this time you get it cut off. Then I went with him to a service, he was really keen and when we went there the lady who was there said, ‘Oh no, I can’t help him’ ….”*
(Service Provider 6, male)

The location of residence and the distance travelled to reach the services were also a problem reported as one of the barriers to accessing MHSs. Both community and service provider participants reported transportation and costs related to reaching the services as causes of underutilisation of the MHSs:


*“The location is also another big barrier (to the access to MHSs). Because a lot of refugees now tend to stay in groups in certain areas. So, if I was to advise the government what is the best thing you want to—I’ll probably always try and put services in the place where these people are so that at least they can access them easily”*
(African migrant 1, male, South Sudan)


*“The barriers are the locations of certain places that people need to travel, not understanding or not knowing how to catch public transport”*
(Service Provider 7, female)


*“Another problem for some people, I think, is transport. Not every family has access to a vehicle and when it comes to such thinking: we’ll catch one bus and then we’ll go and catch another one and that becomes a bit of problem”*
(African migrant 13, male Democratic Republic of Congo)

### 3.3. Structural or Health System-Related Barriers

The stories of African migrants identified health system-related factors that influenced their decision to accessing MHSs. For example, unavailability of quick appointments made it a challenge for people to be served when they needed the MHSs immediately. Similarly, the long waiting time to meet service providers such as a counsellor or general practitioner (GP) poorly influenced access to MHSs:


*“You have to book and then maybe they give you appointment next week and by the time you reach on that day, maybe that feeling which you are having because you’ve lost somebody in your family or this and this and this, maybe it’s already covered. So that’s another barrier that is not actually immediate service which you can get straight away”*
(African migrant 14, male, Burundi)


*“When I was working (role redacted to de-identify the participant) I got a report of the clients who (wanted an) appointment to see doctors, and every time they ring, “Oh, we are fully booked. Maybe call us in two weeks’ time,” and then two of the families go—told me that they went in ED—emergency department, and they were kept waiting for a long time before they come in”*
(African migrant 13, male, Democratic Republic of Congo)

Being served by a different GP each time and rushed service provision by GPs were also factors that influenced access to MHSs among African community members. The following story of a female African migrant who has lived in Australia for 10 years, illustrates the potential impact of having service providers change over time:

Q:
*“It’s really interesting that you’ve said you’d be happy to talk to someone (talk to a counsellor about your mental health issue), but you just don’t know who or where to go to find them. Is that right?*


A:
*Yeah.*


Q:
*Yeah. And your GP hasn’t helped with that, your GP hasn’t helped you connect to someone (e.g., a counsellor)?*


A:*They gave me different GP. Since I’ve been here, I do not have particular one GP. The GP I used to see before, today when you book—you want to go see a GP, they will give you another GP. Then when you book, then they give you another GP. I keep telling I want to continue seeing one single GP. Not everyone you can go to talk to”* (African migrant 12, female, Liberia).

The impact of changes in service provision through providing new or different service providers was also identified by service providers participating in the study as one of the barriers to accessing MHSs by African migrants, as highlighted in this excerpt:


*“I have really discussed with people who refuse to go seeking for help because they are being disappointed. Because they are not being listened to, because they have been rejected, they were not satisfied with the services. …. I had another one who called me. He said to me, “Look can you change my GP?” And I said, “But why should I change your GP?” “Because I went there (to the GP) and I was not satisfied.” She told me, “You have to do quickly, I have a lot of clients waiting there.” …. that is a bit disappointing, so he wants to change that GP because he is not happy with the services”*
(Service Provider 6, male)

### 3.4. Sociocultural Factors

#### 3.4.1. Stigma and Discrimination

Mental Health stigma and shame were also reported by both African migrants from across African countries and service providers as a determinant of poor access to MHSs. Their stories suggested that mental health stigma, social isolation and a fear of being left out or being avoided by other community members led to community members refusing to accept themselves as having the MHPs, which also prevented them from seeking the needed support:


*“I have mental health issue, but I will not admit, or I will not accept that I actually have some mental illness. Because of that fear of the stigma (as crazy persons) from the community, that if I admit and say: I have some mental illness then I will be isolated from the community”*
(African migrant 1, male, South Sudan)


*“I think the stigma and discrimination associated with mental health issue within the communities, it’s quite shunned upon. They’re quite—they don’t want the other community members to know, which—it’s really quite disheartening, but it is what it is. So, I think they’re denying themselves to actually have safe supports from within their community”*
(Service Provider 7, female)

As a consequence, affected individuals were reluctant to talk about mental health issues or disclose their experience of mental health issues, as described in the following narratives of an African migrant who has lived in Australia for 16 years:


*“I don’t want to disclose this, I don’t want to disclose that I’ve been through this. I might be mocked or looked down at. It’s a very small community. My name will be there”*
(African migrant 3, male, Democratic Republic of Congo)

A service provider reflected on this shame and stigma associated with mental health issues, and contrasted this with physical health conditions:


*“It’s a bit shameful if you’ve got a mental illness. It’s not seen as similar to if you had a physical—if you had a physical break that’s okay, but if you have something going on inside your mind that you can’t see, it’s taken more as you’re not right in the head, rather than situational”*
(Service Provider 2, male)

Participants from different African countries shared stories about the impact that mental health stigma had on people’s social life and relationships with families and friends within their communities, and the discrimination they felt. The narratives below of some African participants show their anxieties related to the impacts of mental health stigma, such as people with MHPs being regarded as outcasts, not part of the community, negatively judged and labelled, and what they said would not be listened to or taken seriously by others, due to the perception that they had a mental health problem:


*“There have been a few cases recently where a few girls—who are actually just a few years older than me, they were admitted to mental health institutions because they had a reached a point where they were just keeping it in, keeping it in. They had just broken down completely. And then now there’s this negative connotation attached to them, their family, and then that sort of just shunned a lot of other girls from being able to speak out, because they’re afraid of it impacting the family and seeing how it went”*
(African migrant 9, female, Sierra Leone)


*“You’re not a part of community anymore because you have mental health issue and have been visiting mental health clinic, you know? So, this why people may not like to disclose that. You are considered as an outcast (if you have a mental health issue). People will not listen to you if you have mental health illness. So, whatever good idea you may come up with, people will not consider it. That’s the problem”*
(African migrant 3, male, Democratic Republic of Congo)


*“I got a clear picture when I came to Australia because back home, if there is a mental illness, it’s like, you’ve been written off. They use that to judge you, they label you. Anything you would talk or do, they will not take you seriously and say, ‘he is mad, he has got mental illness, you know?”*
(African migrant 15, male, Sierra Leone)

The fear of stigma and discrimination associated with mental health issues were further detailed as factors that demotivated African community members across different cultures from accessing MHSs. Participants described how people feared attending mental health clinics, GPs or counsellors, wanting to avoid discriminatory and stigmatising attitudes and behaviours of other community members:


*“I think in my country, especially the (tribe’s name redacted for confidentiality), they say if you are seeing a GP or a psychologist, they will think you are crazy. They will be like, you are crazy. And sometimes they say people who are seeing the counsellors and the GP and psychologists are crazy people. …. It (being crazy) means someone with the wrong mentality. Their brain does not have good thoughts, or the person does not act quite good, or has a problem in their brain, or has a mentality problem”*
(African migrant 17, female South Sudan)


*“Even when they come there (mental healthcare facilities or services), they are not free to share much and sometimes they don’t want even to be known that they’ve gone there. They regard that—and this is for many—when they are being known that they are going to see someone to deal with their psychological or mental health, that they’re crazy”*
(African migrant 14, male, Burundi)

#### 3.4.2. Cultural Practices in Spousal Relationships

Cultural practices related to spousal relationships and individual roles within families also seemed to influence the way people access MHSs. For example, cultural practices where husbands were in control over decision making within families and women needed to get permission from their husbands before engaging in any activities were examples of cultural-related barriers to accessing MHSs described by both male and female participants across different countries, including Somalia, South Sudan, Democratic Republic of Congo and Liberia. Illustrations of such cultural practices are provided by the following quotes from some participants, both from the community and service providers:


*“If they are women, women will never, never ever seek for assistance from our community. It’s a strong community that depends on husband’s permission”*
(African migrant 5, male, Democratic Republic of Congo)


*“Another barrier is some of them (women), they are on this mental problem and they are still with their husband. And it’s hard for them to be with their husband and their husband does not want them to do this (access mental health services)”*
(African migrant 16, female, Liberia)


*“Well, because I am finding that particular family that I’ve worked with, the husband is always the talker, and the wife is quite reserved. So, there has been times where I have noticed from a cultural perspective that it’s becoming quite clear that maybe the husband is quite dominant over the female”*
(Service Provider 7, female)

In addition, both service providers and African migrants identified cultural perceptions that characterised men as being strong as a factor that led to poor access to MHSs. It was strongly reported that the African culture dictated that when ‘a man’ was seen to have a MHP and accessed MHSs, this was an indication of him being ‘weak’, and at risk of losing his value (masculinity and power):


*“Most of the Africans are brought up in a very cultural way where you are set up to be tough. You know, as a tough person. So, by saying words like I’m stressed and all this, then they regard, yeah, this man is not a tough guy”*
(African migrant 9, male, Sierra Leone)

It was also noted that African men particularly, seemed to more affected by their cultural perspectives which expected men to be strong and not as weak as women. As such, men feared to talk about MHPs or to disclose having sought mental health service, as they did not want to demonstrate their perceived weakness. These sentiments were pointed out by community participants as well as service providers as follows:


*“It’s something that, it’s cultural. When a guy reveals these things (mental health issues), they usually fear to be judged. They will be like this guy is very weak, and he doesn’t act like a man. So that’s why they usually keep their feelings inside because they are fearing to be judged. It’s a fear they have and also their cultural beliefs, so I think, yeah. They will feel like they’re weak. And they feel like they’re not macho, or they’re not men enough, because they usually say, you shouldn’t hurt a woman; that’s what is usually said. So, they say women heart is weak, and so if the men act, they will be like, don’t act like a woman. Be like ‘a man’. Stop revealing everything”*
(African migrant 17, female, South Sudan)


*“They don’t want to talk about it, because they think they’ll look weak. …. Men, they don’t talk about it at all. Men don’t share emotions. So, it will be almost impossible to bring men to see counsellors. Women you can still try and try and try and you can go, and with their men, oh, my God”*
(Service Provider 1, female)

### 3.5. Religious or Spiritual-Related Perceptions on MHPs

Religious factors were reported, by both the service providers and African migrants (both male and female) from across countries, as playing an important role in influencing people’s access to MHSs. Religious perceptions that MHPs were God’s punishment, spiritual or supranatural evil that cannot be helped, seemed to be an important deterrent to accessing MHSs. These assertions are illustrated by the following participants’ quotes:


*“They will just say—because a lot of people believe in God and people who really don’t have education or so on, they think it’s a God’s punishment. So, if it is God’s punishment, no service can help you. So, I guess they don’t feel good when they think it’s something like this—they need to refer to faith”*
(Service Provider 6, male)


*“Like, mental illnesses are considered in Ethiopia that they are like supernatural, come from the evil, come from God’s punishment. Like people believe in witchcraft and others. So, you know, those things are the things that will hinder or stop people from seeking mental health services here, and like people here, people will trust more to the Church and the priest around them”*
(African migrant 2, male, Ethiopia)

These religious perceptions seemed to have a strong influence in African migrants’ responses to MHPs. Both African migrants and service providers indicated that African migrants responded to MHPs through praying and searching for support from religious leaders such as priests (in Christianity) and Imams (in Islam), rather than accessing MHSs:


*“They will go to the church; they will pray or to the mosque for the Imam to pray for them. So, they will just say; I have nightmares, I feel sad, or I cannot sleep, I cannot eat. Those are sign of depression”*
(Service Provider 6, male)


*“Sometimes they would go with a faith leader. Because a lot of people have this idea that if you are experiencing mental illness, you are cursed, et cetera. And you do need some sort of healing through faith”*
(African migrant 8, female, Somalia)

In line with religious beliefs described above, MHPs were not acknowledged by African migrants (both from their country of origin and in Australia) as medical conditions. As such, it was a common practice for people who suffered MHPs to seek support from religious leaders who delivered non-medical services. These were perceived barriers of access to MHSs described by people from different countries as shown below:


*“Even back home, even now, I don’t think that they (people in Africa) have many programs or types of services like what we have here. So, because of that, many people just rely on family or those who are Christian, maybe rely on churches and ministers for their support. Up to now, that’s still existing that maybe we don’t regard it is an issue”*
(African migrant 14, male, Burundi)

With the western model of MHS delivery perceived as being insufficient to cater form African communities’ MHP needs in Australia, some participants demonstrated the benefits of services provided by religious leaders in addressing MHPs to some of their believers. These assertions are substantiated from the narratives of the following female participant from South Sudanese community:


*“They (religious leaders) come and counsel you, they teach you a lot of work in the Bible, that there are some people who struggle in the Bible and also sometime some people help you. And they can help you to come and do it, to work for you”*
(African migrant 11, female, South Sudan)

## 4. Discussion

Mental health issues are a major public health problem that affect migrant communities, especially those with a refugee background [5,46], and people living with mental issues regardless of their origin often face substantial challenges including the inability to access the needed care and treatment [46,47,48]. As such, available and accessible MHSs are necessary determinants to improve health and well-being and facilitate integration [47]. This study aimed to understand factors that influenced access to MHSs among African migrants in South Australia and identified a range of barriers. These findings are critical because they may indicate the reasons why MHSs are generally poorly accessed by CALD migrants, despite the Australian government promoting the Australian Mental Health Plan that aims to provide general MHSs for all Australians [33].

The current findings suggest that the lack of knowledge and understanding of MHPs alongside inadequacies in availability of MHSs influenced the access to MHSs among Africans, supporting the findings of previous studies [28,34,36]. Of importance, poor mental health literacy seemed to not only hamper the African migrants from recognising, managing and preventing MHPs, but deterred them from seeking help from the currently available support and/or MHSs. These finding are in alignment with concepts of the access to healthcare framework [39], further suggesting that people’s ability to perceive their need for healthcare services is determined by their health literacy, including knowledge about the specific health issue and the availability of healthcare services for them to access [48]. It is therefore reasonable to suggest that poor health literacy, especially regarding availability of MHSs among African migrants, is an indication of poor approachability of the said services, which may relate to limited dissemination of information about the MHSs, and their availability to the African communities. The access to healthcare service framework asserts the importance of the approachability of services and/or how well the services are known to patients/populations and, for the current study, how well the MHSs are known to African migrants is a critical element of accessibility for any health care service [39].

Consistent with the healthcare services framework [39,49,50,51], unaffordability of healthcare services and transportation-related costs due to poor financial conditions were some of the personal barriers to accessing MHSs among African migrants. The lack of perceived benefits of MHSs could also be categorised as one of the barriers that influenced access to the services among African communities. The perceived lack of benefits of MHSs seemed to be based on the participants’ experiences and assessment of the content of the services, which seemed to be perceived as not addressing their problems and were unhelpful, indicative of the failure of MHSs to address the needs of migrant populations. These findings support previous studies and the healthcare service framework concept [28,38,39] that identified the inappropriateness of healthcare services as an important determinant facilitating poor access to services.

### 4.1. Perceptions of MHPs, Health Literacy, Religiosity and Mental Health Stigma

The study findings further revealed that migrants’ conceptualisations of MPHs led them to perceive people with mental health conditions as being crazy or abnormal. This may relate to health literacy and lack of recognition or categorisation of issues such as stress, anxiety and depression as MHPs. As noted above, poor health literacy is a significant predictor of access to health services [49,52], which may be the case for African migrants, especially those with a refugee background, when compared to the general community [48,52,53] in Australia.

In addition to poor literacy, religious and/or spiritual-related perceptions of MHPs seemed to influence access to MHSs among African migrants. This is an important finding, which is different from previous studies on healthcare service utilisation among African communities [11,37,38,49]. The religious or spiritual-related perceptions within African communities that associated MHPs with God’s punishment, supernatural events, and the act of the devil, seemed to significantly influence their health seeking behaviours. Those seeking help relied heavily for mental health cure/treatment and other support on religious leaders such as priests or imams through prayers. In addition to the complex perceptions related to MHPs described above, the trust they had in religious leaders, common practices of seeking support from religious leaders, and the existing practices of receiving socio-medical supports from religious leaders that stemmed from their countries of origin back in Africa, were the factors that supported their choice in seeking religious leaders’ support rather than that of MHSs. The suggests the need for MHSs to work more closely with religious leaders, to not only facilitate access to MHSs but also for MHSs to improve their cultural appropriateness.

Additionally, underpinning fear of disclosure and social isolation was another mental health stigma. As seen in the study’s findings, having MHPs was thought to be shameful and led to being looked down upon and disregarded by people within their communities. Compared to women, African men seemed to be particularly concerned with the possibility of being looked down upon, regarded as outcasts, as not part of the community, not being listened to, or not taken seriously by others due to MHPs. Such concerns seemed to be influenced by the perceptions within African communities that, for ‘a man’ to be diagnosed with a mental health issue was a sign of weakness with a potential to devalue them as ‘a man’. Goffman acknowledges that the fear of being stigmatised, socially isolated and disgraced leads to many aspects of people’s behaviours [54], and it appears that mental health stigma deterred African migrants across different countries from accessing MHSs in South Australia. These findings are similar to other studies that depict stigma, discrimination, and other social factors as important barriers to accessing health services among African and other migrants from other countries [41,55,56]. The negative social consequences associated with stigma and discrimination appeared to demotivate African migrants from engaging in MHSs, which are in line with the concept of ability to engage in services explained in the access to healthcare service framework [39].

### 4.2. Structural Factors including Health System and Patriarchal Beliefs as Barriers

This study presents new cultural-related factors that hampered African migrants from accessing MHSs that have not been explored in previous studies of barriers to healthcare service utilisation among African migrant communities [9,36,37]. For example, the African cultural practices where men or husbands are bestowed with power and deemed to be decision makers within a family/community, including how family members/community access healthcare services, hampered women and other family members’ access to MHSs. The current study suggests that some women in African communities did not feel free to make their own health decisions and may not be permitted by their husbands to access MHSs.

Consistent with previous studies [28,34,37,53], some structural factors, including complex service-related procedures such as an upfront booking appointment, the long waiting list to access services, attitudes and behaviours of service providers, and the regular changes to service providers (i.e., different GPs and/or counsellors), were reported to influence access to MHSs among African migrants. Of note, the unsupportive attitudes of some service providers (such as asking a client to hurry up as other clients are waiting) and the lack of having the same service provider consistently to establish familiarity, influenced their unwillingness to disclose about their MHPs due to feeling rejected or not welcomed. These findings are in alignment with the concept of acceptability in the healthcare services framework, suggesting that healthcare providers’ attitudes and behaviours are important, and do influence accessibility and acceptance by intended service users [30].

In summary, it important to note that the current study findings support several previous studies conducted with a range of CALD migrants, including Middle Eastern, Central Asian, and South Asian refugees in host countries [26,27,57]. The similarities in findings included lack of knowledge about the services, lack of understanding, different conceptualisation of mental health issues, language barriers, social stigma, and lack of trust in service providers [27,35]. For example, a Systematic Review by Byrow et al. [26] examine mental health perceptions and barriers to help-seeking behaviours amongst refugees and noted the consistent findings across culturally diverse groups and the potentially lower acceptance of formal treatment or assistance for mental health issues.

### 4.3. Study Limitations and Strengths

The current study has some limitations. The study involved a small number of male and female African migrants in South Australia, and this could have led to a biased overview of the perceptions and experiences of African migrants about factors that hindered their access to MHSs. Therefore, as is the case for many qualitative studies, the current findings cannot be generalised to all African migrants in other parts of Australia or globally. The distribution of study information packs through organisations or institutions providing MHSs or support for migrant populations, used to recruit service providers, was also a possible limitation as this might have resulted in the recruitment of service providers from similar networks and the absence of the perspectives and experiences of service providers who were not directly involved in MHS delivery to migrant populations. Additionally, the current study was conducted during the COVID-19 pandemic. It is therefore possible that some African migrants who did not have access to digital platforms such as WhatsApp/Zoom, smart phones or computers were excluded from being potential participants. However, the strengths of this study were that it had a clear identified purpose and methodological design. The qualitative design applied in this study helped explore in-depth views and experiences of African migrants and perspectives of services providers regarding the topic under investigation. Another strength of this study was the application of a qualitative data analysis framework to guide the analysis of the study findings, which facilitated the analysis of the data in a coherent and structured way and enhanced transparency, rigour and the validity of the analytic process. In addition, to our knowledge, this is the first qualitative inquiry to combine the views, perspectives and experiences of both African migrants and service providers in the context of Australia exploring access issues to mental health services. The current findings are useful for the development of programs and activities and for the improvement of the mental healthcare system and delivery to address barriers to the access to MHSs among African migrants and other migrant populations.

## 5. Conclusions

This study reports a range of complex determinants that influence access to MHSs among African migrants in South Australia. Personal factors, including the lack of knowledge of MHPs and MHSs, low perceived benefits of MHSs, difficulties related to healthcare cost, transportation cost and systems, and structural factors related to the procedure for accessing MHSs, were common barriers to accessing MHSs among African migrant communities. Sociocultural factors including fear of stigma, discrimination, social isolation and exclusion, avoidance, male domination in family decision making, concern about characteristics of masculinity and power, and religious or spiritual-related perceptions of MHPs as God’s punishment, supernatural events, or the act of the devil, were also barriers. The findings indicate the need for education on mental health problems and services for African migrants to improve their understanding and recognition of the issues and their access to available MHSs. The findings also indicate the need to address sociocultural and religious aspects or practices and perceptions which influence their health seeking behaviours as parts of a mental health education program, which may lead to better understanding and recognition of the influence of these aspects on their health and well-being. Improving MHS provision by providing culturally safe services and consultation with key community leaders, including religious leaders, could be another factor that would improve accessibility of MHSs for these communities. Future studies could particularly investigate the perspectives of community and religious leaders on how to improve understanding of African migrants about MHPs and the available MHSs, as well as further examination of any differences within African communities.

## Figures and Tables

**Figure 1 ijerph-18-08906-f001:**
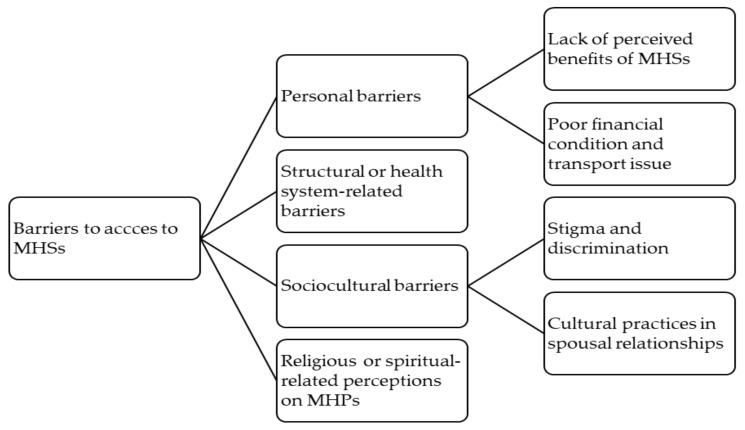
Coding tree with the main themes and sub-themes.

**Table 1 ijerph-18-08906-t001:** Sociodemographic profile of the of participants.

African Migrants	
Characteristics	*N* = 20 (%)
Age ≤30 31–40 41–50 51–60	4 (20) 4 (20) 6 (30) 6 (30)
Country of Origin The Democratic Republic of Congo South Sudan Liberia Sierra Leone Burundi Ethiopia Kenya Somalia	5 (25) 4 (20) 3 (15) 3 (15) 2 (10) 1 (5) 1 (5) 1 (5)
Employment status Employed Unemployed Student	11 (55) 7 (35) 2 (10)
Duration of Living in Australia (year) 1–5 6–10 11–15 16–20	5 (25) 2 (10) 8 (40) 5 (25)
**Service providers**	
**Role**	***N* = 10 (%)**
Counsellor	2 (10)
Clinical services manager	1 (5)
Community development officer	1 (5)
Refugee support program worker	1 (5)
Mental health and well-being manager	1 (5)
Mental health worker	1 (5)
African family support worker	1 (5)
Settlement engagement and support managers	2 (10)

## Data Availability

Data generated or analysed for this study are not available (publically or privately) due to highly sensitive nature of the interview material.

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
