# Peer review of "Migrants and Service Providers’ Perspectives of Barriers to Accessing Mental Health Services in South Australia: A Case of African Migrants with a Refugee Background in South Australia"

_ijerph, 2021, doi:10.3390/ijerph18178906_

Round 1

Reviewer 1 Report

I have read with interest this paper. I believe that the paper is interesting. However, I have some concerns that are reported herein.

  • Introduction: Authors should indicate previous studies about MH in migrants population during COVID 19 pandemic.
  • Material and methods: Theoretical framework don’t should be include in this section. Please move to introduction section.

In addition, Theoretical framework should finish with the aim of the study. Please, include it here.

Data collection procedure : authors should include the interview guide for better understanding.

Data analysis: Have authors use a software for data analysis?

Please provide a coding tree with the main themes.

  • Results: a table with sociodemographic profile should be included.

*Author should use the COREQ checklist and they should include as a annex.

Author Response

Reviewer one

Open Review

    English language and style

( ) Extensive editing of English language and style required
( ) Moderate English changes required
( ) English language and style are fine/minor spell check required
(x) I don't feel qualified to judge about the English language and style

Yes

Can be improved

Must be improved

Not applicable

Does the introduction provide sufficient background and include all relevant references?

( )

( )

(x)

( )

Is the research design appropriate?

( )

( )

(x)

( )

Are the methods adequately described?

( )

( )

(x)

( )

Are the results clearly presented?

( )

(x)

( )

( )

Are the conclusions supported by the results?

( )

(x)

( )

( )

Comments and Suggestions for Authors

I have read with interest this paper. I believe that the paper is interesting. However, I have some concerns that are reported herein.

Introduction: Authors should indicate previous studies about MH in migrants’ population during COVID 19 pandemic.

Response

  • Previous studies about MH in migrant population during COVID-19 pandemic have been added to the first and last paragraphs of the introduction section:
  • The current COVID-19 pandemic is another issue that has also been reported to increase the prevalence of and exacerbate MHPs such as anxiety, depression and stress within migrant and refugee populations in general in Australia and throughout the globe (18-21).
  •  
  • In addition, the current COVID-19 pandemic is also reported as a barrier to the access to MHSs among migrants and refugee populations in Australia and many settings due to worry, stress and fear of exposure to COVID-19 infection, and is considered exacerbating the pre-existing barriers to accessing MHSs among these populations (18-21, 38).
  •  

Material and methods:

Theoretical framework don’t should be include in this section. Please move to introduction section.

Response:

  • Theoretical framework has been moved to introduction section

In addition, Theoretical framework should finish with the aim of the study. Please, include it here.

Response:

  • The aim of the study is now provided at the end of theoretical framework.
  • Guided by the access to healthcare framework, this study aims to explore personal, structural, sociocultural and religious barriers to accessing MHSs among African migrants in South Australia. Understanding these barriers is necessary for governments and policy makers to address them at policy and practical level through programs or interventions that improve African migrants’ access to the services and mental health outcomes.

Data collection procedure: authors should include the interview guide for better understanding.

Response:

  • The interview guide has been provided.

Data analysis: Have authors use a software for data analysis?

Response:

  • Data were analysed thematically using the framework approach, undertaken manually using Microsoft word (version 10) to group themes. The framework approach comprises five steps of qualitative data analysis, including familiarisation with the data, identification of a thematic framework, indexing the data, charting the data and mapping and interpretation of the data. Information about this has been provided in data analysis section.

Please provide a coding tree with the main themes.

Response:

  • A coding tree with the main themes and sub-themes has been provided at the end of results section.

Results: a table with sociodemographic profile should be included.

Response:

  • Table has been provided.

*Author should use the COREQ checklist, and they should include as a annex.

Response:

  • COREQ checklist has been provided.

Submission Date

12 July 2021

Date of this review

27 Jul 2021 18:25:23

Reviewer 2 Report

The paper submitted for review is a qualitative study on the barriers to the use of mental health services among emigrants in Australia.

The concept of research, research methods, analysis and interpretation of results are presented correctly. I have no objections to them.

The only aspect that could be reconsidered is the modification of the introduction. For a better understanding of the situation of migrants and their problems in Australia, quantitative data on the number of migrants and the percentage of migrants from African countries should be provided. What is more, the scale of mental disorders incidence in Australia and healthcare infrastructure is worth presenting. The knowledge about the medical aspect will allow for a better understanding of the study.

Author Response

Reviewer two comments

Open Review

English language and style

( ) Extensive editing of English language and style required
( ) Moderate English changes required
( ) English language and style are fine/minor spell check required
(x) I don't feel qualified to judge about the English language and style

Yes

Can be improved

Must be improved

Not applicable

Does the introduction provide sufficient background and include all relevant references?

( )

( )

(x)

( )

Is the research design appropriate?

(x)

( )

( )

( )

Are the methods adequately described?

(x)

( )

( )

( )

Are the results clearly presented?

(x)

( )

( )

( )

Are the conclusions supported by the results?

(x)

( )

( )

( )

Comments and Suggestions for Authors

The paper submitted for review is a qualitative study on the barriers to the use of mental health services among emigrants in Australia.

The concept of research, research methods, analysis and interpretation of results are presented correctly. I have no objections to them.

  • The only aspect that could be reconsidered is the modification of the introduction. For a better understanding of the situation of migrants and their problems in Australia, quantitative data on the number of migrants and the percentage of migrants from African countries should be provided.

Response :

Thank you for your feedback. The introduction has been modified and the following information has been added.

“Australia has a significant migrant population, with arrivals from many developing settings including from Asia and Africa (1, 2). The 2020 Australian Bureau of Statistics report showed that there were over 7.6 million migrants living in Australia, of whom 380,000 were from African countries (3). In terms of migrants with refugee background, Australia recognised or resettled 180,788 refugees during the period from 2009 to 2018, which was 0.89% of an estimated 20.3 million refugees globally (4). For African migrants, the majority have migrated to Australia as refugees due to a wide range of factors including civil conflicts and/or war in their home countries (5, 6). They may have had negative experiences during relocation, especially when residing in refugee camps, and resettlement challenges make it more complex to settle in host countries (6-9). In Australia, African migrants have been reported to experience a disproportionate burden of health problems related to postmigration resettlement, such as trauma, separation from family and peer networks, difficulties in social adaptation and adjustment to new systems, unemployment, and challenges accessing housing and education (5, 7, 10-12). Such issues facing African migrants, particularly those arriving with a refugee background can cause mental health problems (MHPs) such as depression, schizophrenia and stress conditions (7, 13-15), further affecting resettlement processes and engagement with mental healthcare services (MHSs) in Australia (5, 9, 16, 17). The current COVID-19 pandemic is another issue that has also been reported to increase and exacerbate MHPs such as anxiety, depression and stress within migrant populations in general in Australia (18-21). MHPs within migrant populations, including African migrants seem to add up to the overall increase of the prevalence of mental disorders in Australia. The 2020 Australia’s Health report showed that one in five Australians (20.1%) experienced a mental health condition in 2017-2018, an increase from 17.5% in 2014-2015 (22)”.

What is more, the scale of mental disorders incidence in Australia and healthcare infrastructure is worth presenting. The knowledge about the medical aspect will allow for a better understanding of the study.

Response:

 We have added in general terms,  the prevalence of mental health disorder related to migrants with a refugee background as will related to most Africans resettled in Australia due to them having similar backgrounds

Submission Date

12 July 2021

Date of this review

09 Aug 2021 08:57:46

APPENDIX 1 CONSOLIDATED CRITERIA FOR REPORTING QUALITATIVE STUDIES (COREQ): 32-ITEM CHECKLIST

No 

Item     

Guide questions/description 

Page

Domain 1: Research team and reflexivity 

Personal Characteristics 

1.

Interviewer/facilitator

Which author/s conducted the interview or focus group?

4

2

Credentials

What were the researcher's credentials? E.g. PhD, MD 

4

3

Occupation 

What was their occupation at the time of the study? 

4

4

Gender 

Was the researcher male or female? 

4

5

Experience and training

What experience or training did the researcher have? 

4

Relationship with participants    

6

Relationship established 

Was a relationship established prior to study commencement?

4

7

Participant knowledge of the interviewer 

What did the participants know about the researcher? e.g. personal goals, reasons for doing the research 

5

8

Interviewer characteristics 

What characteristics were reported about the interviewer/facilitator? e.g. Bias, assumptions, reasons and interests in the research topic 

5

Domain 2: study design 

Theoretical framework    

9

Methodological orientation and Theory 

What methodological orientation was stated to underpin the study? e.g. grounded theory, discourse analysis, ethnography, phenomenology, content analysis 

2-3

Participant selection 

10

Sampling

How were participants selected? e.g. purposive, convenience, consecutive, snowball 

4

11

Method of approach 

How were participants approached? e.g. face-to-face, telephone, mail, email 

4

12

Sample size 

How many participants were in the study? 

4

13

Non-participation 

How many people refused to participate or dropped out? Reasons? 

4

Setting 

14

Setting of data collection 

Where was the data collected? e.g. home, clinic, workplace 

4

15

Presence of non-participants 

Was anyone else present besides the participants and researchers? 

4

16

Description of sample 

What are the important characteristics of the sample? e.g. demographic data, date 

5

Data collection 

17

Interview guide 

Were questions, prompts, guides provided by the authors? Was it pilot tested? 

4

18

Repeat interviews            

Were repeat interviews carried out? If yes, how many? 

4

19

Audio/visual recording            

Did the research use audio or visual recording to collect the data? 

4

20

Field notes       

Were field notes made during and/or after the interview or focus group? 

4

21

Duration           

What was the duration of the interviews or focus group? 

4

22

Data saturation 

Was data saturation discussed?

4

23

Transcripts returned            

Were transcripts returned to participants for comment and/or correction? 

4

Domain 3: analysis and findings

Data analysis   

24

Number of data coders 

How many data coders coded the data? 

4-5

25

Description of the coding tree 

Did authors provide a description of the coding tree? 

13

26

Derivation of themes 

Were themes identified in advance or derived from the data?

4-5

27

Software

What software, if applicable, was used to manage the data? 

5

28

Participant checking 

Did participants provide feedback on the findings? 

4

Reporting         

29

Quotations presented 

Were participant quotations presented to illustrate the themes / findings? Was each quotation identified? e.g. participant number 

6-12

30

Data and findings consistent 

Was there consistency between the data presented and the findings? 

6-12

31

Clarity of major themes 

Were major themes clearly presented in the findings? 

6-12

32

Clarity of minor themes 

Is there a description of diverse cases or discussion of minor themes? 

6-12

APPENDIX 2: INTERVIEW GUIDE

African migrant participants

What do you know mental health problems? Please explain.

  • How mental health problems are perceived in your communities?
  • What factors do you think influence your perceptions or perceptions in African communities about mental health problems?
    • Social, cultural and religious factors? Why? Please explain.
    • How they influence your perceptions about mental health problems?
  • Do people talk about mental health problems in your communities? Why? Please explain.

What do you know about mental healthcare services? Please explain

  • Where are the services available?
    • What kind of services have you accessed?
    • What are the procedures to access the services? Are they easy and helpful? Why?
    • How do feel about the services you have received? Comfortable or not and why?
  • Do you think the services meet your expectations? Why? Please explain
  • What do you think about benefits of the services to you? Please explain.
  • What do you think about the influence of community perceptions about mental health problems on your access to mental healthcare services? Pleases explain.
    • Do you think such perceptions also influence the access of African migrants to mental healthcare service? Why? Please explain.

What about the decision in your family about access to mental healthcare services?

  • Who makes the decision? Why?
  • What factors do you think influence your or your family decision to access or not to access mental healthcare services? Why? Please explain.

Do you think the perceptions of mental health problems in your culture or religion influence your access or the access of African migrants to mental healthcare services?

  • How and why? Please explain more about these.

Are there any costs related to the access to mental healthcare services?

  • What are the costs for? Please explain.
  • How do you reach healthcare facilities to access mental healthcare services?
  • How far is it to reach mental healthcare services?

Interview guide for Service providers

What kinds of services does your institution provide for migrant population in South Australia? Please explain.

Based on your experience, what do you think about the level of understanding or knowledge of African migrants about mental health problems? Please explain.

  • They have good knowledge about mental health problems or not? Please explain more about it.
  • What are the perceptions, if any, they have about mental health problems? Why?
  • What influences their perceptions about the problem?
    • Culture, religion? Why?

What do you think about the knowledge or understanding of African migrants about mental healthcare services? Please explain.

  • Do you think they understand the procedures to access the services or not? Are they easy and helpful? Why? Please explain more about these.

Are there any costs for African migrants related to their access to mental healthcare services? Please explain

  • What are the costs for? Please explain.

How do they reach healthcare facilities to access mental healthcare services?

  • Do you think transportation and location where they live have an influence on their access to mental healthcare services? Why? Please explain.

Do you think cultural or religious related perceptions about mental health problems, if any, influence African migrants’ access to mental healthcare services? Why? Pleases explain.

Based on your experiences, are there any cultural related factors that you think influence individual or family decision of African migrants (both women and men or husbands and wives) and families to not access mental healthcare services?

  • What are they? Why? Please explain.

Round 2

Reviewer 1 Report

All questions have been responded. The manuscript have improve and now can be published.